# Hira Makes a Sound: Sustaining High-Impact AANAPISI Innovation in an Asian American Studies Environment before and beyond the COVID-19 Anti-Asian Hate Pandemic

Peter Nien-chu Kiang *, Shirley Suet-ling Tang, Kim Soun Ty, Parmita Gurung, Ammany Ty and Nia Duong

Asian American Studies Program, University of Massachusetts Boston, Boston, MA 02125, USA
* Correspondence: peter.kiang@umb.edu

**Abstract:** This article first describes two high-impact, foundational examples in Asian American Studies over three decades that successfully established and sustained inclusive and equitable educational environments at an urban, public, and federally designated Asian American and Native American Pacific Islander Institution (AANAPISI) research university. Secondly, the article introduces the purpose, process, and product of a fresh programmatic example of cross-generational, community-centered storytelling initiated during the contemporary dual-pandemic period of COVID-19 and anti-Asian hate. Through these examples, we argue that AANAPISIs with longstanding Asian American Studies curricular and pedagogical commitments to transformative educational praxis can offer especially valuable insights for U.S. higher education precisely because the purposeful, strategic attention to developing equitable, inclusive learning environments is so well-developed. Within our own context, we specifically highlight the impacts and importance of developing long-term ecologies to support culturally sustaining curricula and storytelling co-production processes led by core faculty with students and alumni. These examples are particularly salient for under-resourced, predominantly commuter institutions where student engagement with faculty and peers in classroom environments is so vital.

**Keywords:** Asian American studies; AANAPISI MSI; storytelling; curriculum and pedagogy; Nepali diasporic community; anti-Asian hate; Southeast Asian Americans; urban universities

## 1. Introduction

As a distinctive sector of U.S. higher education within the universe of federally designated minority-serving institutions, Asian American and Native American Pacific Islander Institutions (AANAPISIs) represent only 6% of all colleges and universities in the U.S. but enroll 43% of the nation's Asian American and Pacific Islander undergraduate students [1,2]. The University of Massachusetts Boston, for example, is the only majority-students-of-color AANAPISI research university in metro Boston, a region renowned for its uniquely high concentration of wealthy, elite private universities, none of which qualify for AANAPISI status due to the upper-income levels of their students. In contrast, UMass Boston serves local, historically significant Chinese, Vietnamese, and Khmer American communities and other diverse low-income students and families, most of whom are first-generation students of color with financial needs who work and commute. As such, the experiences, perspectives, and profiles of UMass Boston students reflect the racial, cultural, linguistic, and economic realities and recent histories of metro Boston's communities of color. Strategies and practices that create equitable classroom and campus environments for these student populations invariably also lead to longer-term, post-graduation effects of engagement and achievement for alumni who represent and continue to build critical capacities for their families and communities after college.

Despite the minority-serving sectoral significance of AANAPISIs in providing higher education access and support to low-income, underserved Asian American and Pacific

Islander populations, only a few AANAPISIs, such as Sacramento State University, DeAnza College, Hunter College, San Francisco State University, University of California, Irvine, University of Hawaiʻi at Manoa and UMass Boston have historically housed robust Asian American Studies academic units that structurally and culturally provide curricular and pedagogical environments for transformative student learning prior to the launch of AANAPISI Program federal funding streams in 2008. Asian American Studies academic unit-based researchers and practitioners who are grounded within these long-term, high-capacity curricular-rich AANAPISI colleges and universities play key roles in the design, development, sustenance, documentation, and grounded theorizing of effective educational and sociocultural environments for low-income, first-generation and immigrant/refugee students of color and other under-served populations. Such sustained Asian American Studies praxis in AANAPISI contexts should be more fully recognized in relation to the scholarship of equity and innovation in U.S. higher education [3–6].

This article—mindful of three decades of continually reflective programmatic practice, knowledge-building, and synthesis at our own urban, public AANAPISI research university—briefly introduces two longstanding, foundational examples of high-impact intervention in our efforts to establish inclusive and equitable environments through Asian American Studies courses. These two examples, led by co-authors Kiang and Tang, have directly involved co-authors K. Ty, Gurung, A. Ty, and Dương, previously as former students and currently as alumni co-producer/researcher colleagues. This article then turns to focus on a fresh example led by co-authors K. Ty, Gurung, A. Ty, and Dương under Tang's supervision, which highlights new sources of narrative inspiration and impact through a creative co-production process initiated during the contemporary dual-pandemic period defined by COVID-19 and anti-Asian hate. Lessons, implications, and questions emerging from this programmatic innovation and its environmental context then follow. In this way, rather than prioritizing the study of shorter-term interventions or discrete projects, we suggest the importance of longer-term research and documentation to understand how transformative educational environments are not simply developed but programmatically sustained within universities, particularly among minority-serving institutions [3,7].

## 2. Program-Based Classroom and Curricular Environments of Inclusion and Equity

The origin story of UMass Boston's Asian American Studies Program during the 1980s is nationally unique among ethnic studies academic units due to the foundational roles of local Vietnamese and Cambodian refugee students, side-by-side with diverse student veterans from the Vietnam War era. Together, they comprised the program's initial core student constituencies as they enrolled in large numbers with both parallel and intersecting intentions of healing and rebuilding post-war lives through public higher education. Their remarkable confluence in our early Asian American Studies classroom environments inspired powerful curricular and pedagogical purposes that distinctly diverged from the primary focus on racial/cultural identity, representation, and affirmation that has historically and still currently characterized the development of Asian American Studies in other campus settings [8,9]. From this base, yet still influenced by the founding revolutionary purposes of the Asian American Studies field to transform schools and society, we have introduced, developed, and adapted many examples of programmatic praxis to address educational and economic inequities as well as community erasure and resilience amidst the daily life realities of multiracial, multigenerational demographic and cultural changes in our public, urban environment [10–15].

The first following example demonstrates an ongoing programmatic commitment that has simultaneously addressed two basic challenges facing marginalized populations in higher education, particularly in AANAPISI contexts:

- First-generation student of color educational success has a strong positive association with structured, long-term access to both culturally sustaining curricular content/pedagogy and faculty mentors who reflect students' family and community contexts;

- Disaggregated educational data clearly show high needs and under-served, under-resourced realities of Vietnamese, Khmer, Hmong, and other Southeast Asian populations within the Asian American umbrella category, both in K-12 and higher education, locally and nationally.

*2.1. Environmental Example 1: Culturally Specific Curricular Portals*

2.1.1. Southeast Asian Refugee Curricular Interventions Prior to Data Disaggregation Demands

Although the majority of 20+ courses offered by our Asian American Studies Program include some readings, case studies, or projects focusing on Southeast Asian American issues, we have specifically offered Southeast Asians in the U.S. (AsAmSt 225) nearly every semester since originally creating the course in 1989. For over three decades, then, this stable yet ever-evolving course has nationally and locally examined the processes of migration, refugee resettlement, and community development for Vietnamese, Lao, Khmer, and Hmong, with the interplay of themes of trauma, healing, and resilience well-grounded within changing contexts of families, communities, schools, public policies, and homeland relations.

As one of the first institutionalized, semester-long classes in the country developed with this focus, our Southeast Asians in the U.S. course has been offered over 40 times since 1989, reaching ~1200 diverse urban students, including co-authors K. Ty, Gurung, A. Ty, and Dương. Roughly half have been Vietnamese, Cambodian, and ethnic Chinese from Southeast Asia, along with smaller numbers of Lao, Hmong, and others consistent with their community profiles in metro Boston. In the early 1990s, most Southeast Asian students taking the class were second-wave refugees with direct memories of war and refugee flight; in the 2000s, some students were U.S.-born, some were babies during the second-wave migrations, and others came as teens from Vietnam with their fathers and families during the mid-1990s through the Humanitarian Operation program. Currently, in the 2020s, some are from new waves of immigration, many are second-generation, and some represent the emerging third-generation with grandparents who migrated as refugees. Classroom-based research with Southeast Asian American students taking the Southeast Asians in the US course has offered richly grounded perspectives that have informed the development of culturally responsive curricular and pedagogical examples in the fields of education and Asian American Studies, including early articulations of trauma-informed teaching and learning articulated in the 2000s [16,17] and an intersectional, multidimensional conceptual framework used since the 1990s for analyzing the strengths and needs of students as Southeast Asians (Vietnamese, Khmer, etc.), as refugees, as immigrants and as racial minorities [18].

With dramatic changes throughout the past 35 years, not only in the demographic profiles of Southeast Asian American populations but also in the underlying socioeconomic and geopolitical conditions within and between the U.S. and homeland countries, we have frequently updated the AsAmSt 225 course in order to maintain relevance. Opportunities for family reconnection and healing were far fewer, for example, during the early 1990s when return travel to home countries was infrequent, if not dangerous, for refugees, and before global digital telecommunications were readily accessible. Furthermore, the post-9/11 U.S. war on terror during the 2000s infused immediate urgency into the content and conduct of the course because severe lessons from U.S. wars in Southeast Asia were so directly applicable. Additionally, a new generation of refugee students and student veterans were enrolling in our classrooms. In their profoundly ironic course projects, for example, some U.S.-born Cambodian and Vietnamese American student veterans who survived combat in Iraq and Afghanistan revealed how their refugee parents—who had fled from war themselves—had opposed their children's enlistments and prayed throughout their deployments, helplessly waiting for them to return home safely [11,19].

One focus of consistent curricular content in AsAmSt 225 that has remained relevant throughout the entire period of our teaching since 1989, however, has been showing the

realities, causes, and consequences of anti-Asian violence in U.S. society. Actually, one purpose of establishing the course was to provide clarity regarding stories of Southeast Asian refugees, the policies and practices of refugee resettlement through the complex intersections of race, poverty, and language/culture/religion, and the processes needed to nurture healthy Southeast Asian American community development and empowerment over time. These reference points provided explicit contexts for students to understand and challenge the dynamics of intense anti-Asian violence from the 1980s–1990s, which had especially targeted Southeast Asians in metro Boston. In 1987, for example, the year Kiang began designing the AsAmSt 225 course for university approval, 10% of the victims of reported racial violence in the city of Boston were Asian American, even though Asian Americans accounted for only 3% of the city's total population [20]. This level of violence, including several murders, house arsons, car firebombings, and stabbing attacks, along with other forms of harassment and assault during the late 1980s against Southeast Asian refugees, was more frequent and widespread than the explosive, well-documented period of COVID-19 anti-Asian hate currently in the 2020s. The AsAmSt 225 course—from its inception through to the present—has served as a constant source of direct remembrance testimony and documentation, together with critical social/historical/cultural/political analysis, to show how deadly and dangerous are the racist rhetoric, policies and practices of U.S. wars that both target and dehumanize Asians as subhuman enemies, as was the case in Southeast Asia ordered by those at the apex of U.S. military command and control. [21–23].

Though primarily taught by Kiang and Tang, the course has also provided important teaching opportunities for Southeast Asian American community practitioners and graduate students, including a former coordinator of the Massachusetts Office for Refugee Resettlement (Chinese Cambodian), a former undergraduate who became the first Vietnamese American to receive a doctorate in education from Harvard, and UMass Boston's first Hmong American Ph.D. Lao, Khmer, Hmong, and Vietnamese American graduate students and advanced undergraduates have also regularly served in the course as teaching assistants and later became community advocates and bilingual educators themselves. Thus, the Southeast Asians in the U.S. course has served not only as a content- and process-based, high-impact curricular and pedagogical intervention for undergraduate students in relation to their families and communities but also as an ongoing opportunity for culturally responsive mentoring of advanced degree-seeking refugee-background students, prior to the dynamic emergence of younger-generation, U.S.-born Southeast Asian Americans gaining Ph.D.'s and faculty positions at institutions throughout the country during the past decade.

2.1.2. Vietnamese American Studies Commitments

Furthermore, twenty years ago, we developed two additional ethnic-specific, portal-purposed courses to complement the Southeast Asians in the US course. Resources for Vietnamese American Studies (AsAmSt 294), for example, was co-created in 2002 by Kiang with community practitioner/scholar James Bui to explore ways of studying the reconstructions of identity, culture, and community for Vietnamese in the United States and their diasporic relationships around the world. By that time, Boston's Vietnamese refugee/immigrant community was already well-established within the city's Dorchester-Fields Corner neighborhood, just five minutes from the UMass Boston campus. As one of the most vibrant Vietnamese American communities in the U.S. during the past three decades, the neighborhood is also formally recognized by the city and state governments as the Boston Little Saigon cultural district [24–26]. At the same time, consistent with low-income AANAPISI student profiles, the poverty rate for Asian Americans in Dorchester, three-fourths of whom are Vietnamese American, is 27% [27].

Typically taught by a bilingual community practitioner or advanced graduate student, the course featured presentations by local Vietnamese American researchers, writers, community leaders, and visiting scholars. In 2007, the course was embedded with distance-learning and service-learning capacities to support Vietnamese community rebuilding

in the post-Katrina Gulf Coast settings of New Orleans East, Louisiana, and East Biloxi, Mississippi, with 22 students directly contributing to research, organizing, and planning in support of projects related to environmental justice, business and real estate development, community gardening, and advocacy. The cultural responsiveness within the Vietnamese American Studies course environment between the classroom and community leveraged and modeled reciprocal engagements between Vietnamese diasporic communities in Boston and the Gulf Coast during the post-Katrina years [28,29]. Fifteen years later, Vietnamese American undergraduate students currently number over 600 and comprise roughly one-third of all Asian American undergraduates at our AANAPISI university. A Vietnamese American student who took AsAmSt 294 in Spring 2022 with a bilingual Vietnamese American doctoral student instructor reflected:

> *"I think it's an amazing opportunity to be a part of a class dedicated to Vietnamese American Studies—a few years ago, I never imagined I'd be able to take a course like this, but here I am! I feel even more thankful that the class is led by a bilingual Vietnamese American instructor, because her knowledge, experiences, and fluency contribute to an enriching, multilayered, and personal learning experience. As a Vietnamese American student in this class, my culture, family history, spirituality, and language feel validated— at the same time, I am constantly learning and discovering new things about them! ... To have faculty that shares our background and our language offers a connection and understanding that enriches our experience at UMB on a whole other level. This style of teaching and learning is part of what makes the Asian American Studies courses so unique and special to so many students."*

### 2.1.3. Cambodian American Studies Contributions

A parallel course titled "Cambodian American Culture and Communities" (AsAmSt 270) was similarly co-created in 2003 by Tang and Kiang with community practitioner/scholar Sody Lay based on this rationale:

> *"Development of this course recognizes the emergence of the Cambodian American Studies field and the realities of local Cambodian communities in both Lowell and Lynn, Massachusetts—the second and fifth largest concentrations of Cambodians in the U.S. These communities have been sites of research for faculty and graduate students as well as being home to many of our Cambodian American undergraduates [30]. Until now, however, there has been no course at UMass Boston or anywhere outside of California dedicated to the interdisciplinary study of Cambodian Americans. In addition to being open and accessible to all interested students, this course is also intended to serve as a curricular intervention to attract and engage Cambodian Americans at UMass Boston— in light of the reality that Cambodian Americans have the lowest college-going and college graduation rates of any Asian American population within all higher education institutions in Massachusetts."*

Though taught by Tang and others numerous times since its original approval, it took 15 years before co-author K. Ty, who took the course as an undergraduate in 2012, was able to offer AsAmSt 270 as a homegrown instructor after completing her first graduate degree [30]. Since then, she has been the only Khmer American university instructor in the U.S. to teach a Cambodian American Studies course every year for the past six years. The course has attracted Khmer American students of varied backgrounds, including, for example, a third-generation Khmer American student in Spring 2022 whose photo essay project wove together stories related to the impacts of immigration on his personal and family experiences with poverty and domestic violence. While the details of his documentation revealed Khmer refugee realities, such as earning USD 2.00 per day sewing, gang members gathering in his living room, and more, he ended with a clear message for his classmates with echoes for the future: "No matter how hard a situation is, or may seem, there is always a way out."

Importantly, AsAmSt 270 and the other two courses described above have systematically and sustainably reached, motivated, and moved under-represented, under-resourced Southeast Asian American student populations for more than two decades. These ecological effects ensured by secure, well-structured curricular and pedagogical environments have provided significant answers to the urgent question that is rarely articulated alongside data disaggregation demands for Southeast Americans in U.S. higher education: what should be specifically enacted differently? Indeed, if without visions and models of viable, alternative, and long-term learning environments, then simply demanding access to disaggregated data does little to change structures and cultures of educational inequality facing Khmer, Vietnamese, and other Southeast Asian American students, families, and communities [31]. Furthermore, within our own urban public university environment, other diverse students with more recent refugee backgrounds from Sudan, Haiti, Guatemala, Lebanon, Iraq, Afghanistan, and elsewhere have also meaningfully engaged with these Asian American Studies curricular portals. For example, one student majoring in Africana Studies who had come with his family to the United States from Somalia just a few years earlier took Cambodian American Culture and Community to deepen his personal, comparative and applied understandings of refugee realities. After being inspired by the AsAmSt 270 learning environment, he then conceptualized a draft syllabus for a future Somali American Studies course through an independent study of Asian American Studies with Tang and Kiang [16].

*2.2. Environmental Example 2: Storytelling Co-Production and Narrative Wealth Generation*

As a second example of long-term, programmatic purpose and high-impact practice through Asian American Studies at our AANAPISI university, we highlight another twenty-year curricular and pedagogical commitment through which a sustainable educational environment for student- and community-centered story-sharing, story data co-production, and narrative wealth generation has thrived. Comparable to the curricular portal examples above, our storytelling co-production commitments have similarly evolved from the structure and culture of a single, semester-based course—Asian American Media Literacy (AsAmSt 370)—regularly taught by Tang since 2005.

Though originally designed to enable students to become more critically aware of mainstream-dominant stereotypes and the power of independent media representation, Tang transformed the course to align directly with students' urban, working-class backgrounds and their immigrant/refugee families' contexts, assets, and needs [32]. Through a highly structured pedagogy and a curriculum grounded in culturally sustaining, decolonizing, anti-racist frameworks, the AsAmSt 370 course environment has enabled students to take responsibility for creating and sharing their own stories to be producers, not just consumers, of knowledge [33].

Since 2005, we have generated an original archive of ~400 digital stories, most of which are 5–6 min in length, co-produced within the AsAmSt 370 course environment led by Tang. Students' narratives reveal critical themes, from family migration and poverty to health disparities and language barriers, racist violence, gendered discrimination, and more. At the same time, their stories also clearly document authentic resilience, strength, and inspiration [34,35].

While everyone has stories, not everyone has structured opportunities to fully explore, express, and share their stories with purpose and care. In contrast to the small percentage of Asian American students from elite, private universities whose perspectives typically receive far greater attention in national higher education discourse, the modest-income, urban, first-generation Asian American students attending UMass Boston and other public two- and four-year AANAPISIs represent a far greater number and percentage in U.S. higher education overall. Although AANAPISI students may lack the legacy wealth represented by property, inheritance, and college-educated parent networks, they can claim a richness in life stories. Asian American "narrative plenitude" [36] aptly describes our own AsAmSt classrooms every year. The work of uncovering such riches and investing

further in their conversion to narrative wealth, however, does require a secure curricular environment supported by a sustained and powerful pedagogical praxis.

Thus, we are committed to what Tang refers to as textured, layered, purposeful storytelling. This approach links the conventions of academic expectations with the urgent need to address crucial knowledge gaps in research about underrepresented and underserved populations, demonstrating that the institutional challenge of integrating knowledge production and student success at an AANAPISI research university calls for methodological, curricular, and epistemological innovation simultaneously [33]. AANAPISI students must become effective, socially responsible digital storytellers and knowledge producers. Through the course and related initiatives, students learn to center the perspectives, voices, and lived experiences of excluded or marginalized populations, including, but not limited to, just themselves. Students turn to personal narratives to deconstruct and reconstruct the norms established by dominant narratives, not just by adding their own individual faces and cultural settings to diversify the demographic landscape but by using categories of analysis including race, language, legal status, gender, sexuality, socioeconomic status, religious orientations, and other critical themes in Asian American Studies to clarify how historical and social contexts shape their current family conditions and life chances. Students' digital story products are, therefore, intensely personal, political, and positional. This differs from decontextualized, voyeuristic, or aggregated statistical frameworks of data collection and analysis about students of color and their communities. We embrace and affirm students' own family cultural assets, native languages, and ways of knowing.

This curricular and pedagogical environment for storytelling and narrative wealth generation enables students to trust their own voices, develop critical awareness, and embrace their power. Gaining a contextual understanding is important for students to become more fully engaged educationally and civically. This environment also builds community and a sense of belonging because urban students from diverse backgrounds are able to reach each other's common humanity, but only after going deeply into the personal, unique, and private. We cultivate intellectual and emotional capacities by using a mind-body-heart-soul framework that emphasizes wholeness, the ability to firmly hold narrative fragments, and find coherence and meaning in our stories.

Just as they each took the AsAmSt 225 Southeast Asians in the US course as undergraduates with either Kiang or Tang, as noted above, K. Ty, Gurung, A. Ty, and Dương also each took AsAmSt 370 with Tang between 2012 and 2016. Following their respective graduations, each continued their individual involvements as co-production staff in the Asian American Studies Program under Tang, supported by a variety of funding sources. The ongoing, multi-year engagements of these first-generation Khmer, Nepali, and Vietnamese American alumnae have expanded and diversified our available wealth of aesthetic visions, cultural groundings, and community relationships while ensuring a high level of sustained process/product capacity for storytelling. The programmatically strategic role and impact of involving alumni as directly engaged equity assets in AANAPISI contexts have not been previously identified or studied as an effective element or promising practice that enables diverse populations to thrive within inclusive educational environments. Interestingly, while getting through an "ecology of survive" in college typically leads graduating students to quickly disengage as alumni, experiencing an "ecology of thrive" leads alumni to remain engaged in multiple ways, reinforcing and replenishing the thriving ecosystem which they wish to sustain and further advance. For students, faculty, and alumni involved with the organically thriving ecology of AsAmSt 370, the intentional investing of high capacity in order to co-produce powerful real life real stories effectively yields a high impact, which, in turn, creates further capacity to reinvest in further fresh, real-life real-story co-productions.

As such, a notable effect of the AsAmSt 370 course environment has been the emergence and evolution of a deeper, wider platform led by faculty with alumnae/former students serving as staff co-producers and team members. The recruitment, development, and contributions of a collective co-production team of AsAmSt alumnae, in particular, represents a critically important, women-centered, capacity-building innovation with a

profound programmatic impact. In addition to generating an ever-growing core collection of personal narrative videos, we have also developed community documentation projects, multimedia zines, an Instagram site where we collaborate with alumni and other BIPOC artists and writers to explore fresh ways of storytelling, and multiple collaborations with other minority-serving higher education institutions and BIPOC community partners. AsAmSt story products are regularly used for training and education, community dialogues, public policy awareness, and ongoing AANAPISI-focused research. Several have won awards in local film festivals.

In the next section, our focus turns to a newly created storybook, *Hira Makes a Sound*, based on a grounded, community-engaged, co-production process by Tang, K. Ty, Gurung, A. Ty, and Dương intended to address the dual pandemics of COVID-19 and anti-Asian hate in contemporary U.S. society. This work exemplifies what is possible within a narrative-rich, high-capacity, student- and community-centered Asian American Studies environment.

## 3. Results from the Co-Production Environment for *Hira Makes a Sound*

*Bajei (grandma) lost her spirit trying to protect me! If I didn't yell back, this wouldn't have happened!*

When the 2020 pandemic hits, ten-year-old Hira's world is turned upside down. Not only must Hira and her family resist the contagious COVID-19 virus, but they also must face anti-Asian racism. After a violent confrontation outside in their neighborhood, Hira's injured grandmother Ratna Bajei, an immigrant from Nepal, loses her spirit. Hira must find a way to help guide Ratna Bajei home. In the process, she learns to more fully appreciate the importance of the local Nepali community center and her family's cultural practices, as well as the strengths and limitations of her own developing voice.

In early 2021, the digital storytelling team (Tang, K. Ty, Gurung, A. Ty, and Dương) began developing a multilingual children's storybook that would directly address the double-pandemic of COVID-19 and anti-Asian racism. As a team of Asian American women with refugee/immigrant family backgrounds, our shared outrage and grief deepened our determination to complete this storybook project with even greater urgency following the ruthless murders of six Asian and Asian American women frontline workers by a white male shooter in metro Atlanta in March 2021. Unlike course-based co-production projects which focus on students' stories in Tang's AsAmSt 370 course, this project was our own co-creation based on family conversations, community knowledge, and shared narrative imagination. Neither the process nor product of *Hira Makes a Sound* would have been possible without the programmatic environment of relational trust and co-production capacity that we have cultivated and directly sustained over time.

Completed over an 18-month period, the original narrative text and visual content of *Hira Makes a Sound* highlights the importance of sounds and connections within our external and internal environments. Our two main characters respond to anti-Asian harassment using sounds in different ways. Hira is a boisterous and fiery girl with a strong sense of justice. Hira believes in doing what is right, but she is still learning how to use her voice in thoughtful and intentional ways. With wisdom and maturity, Hira's grandmother, Ratna Bajei, believes in using not only her voice but other sounds of being to call for justice.

We created the characters of Hira Gurung and her family to respond, in part, to gaps in anti-racist, multicultural children's literature. Specifically, our review of Asian American children's books revealed an obvious dearth of Nepali American-centered narratives. We took care and responsibility, therefore, to highlight the perspectives of an under-represented Asian ethnic group while showing significant dual-pandemic experiences. Furthermore, we multidimensionally represent Asian elders as precious cultural and knowledge bearers, role models, and activists in our communities. In assessing numerous news stories about recent cases of anti-Asian racist violence, we found elders frequently portrayed as vulnerable, one-dimensional victims. In contrast, our own oral history conversations conducted with local Asian American women elders during the summer of 2021 revealed stories of resistance,

strength, strategy, and deep intuitive knowledge, superpowers enacted in the mundane of daily life.

### 3.1. Research and Documentation

To create *Hira Makes a Sound*, we first envisioned a storybook that is multilingual and intergenerational in its message with a foregrounding of immigrant/refugee children and elders who might be visible targets of racist harassment. A motivating baseline source of story data came through our own family experiences with the double pandemic. Each team member additionally gathered relevant stories and specific oral histories through native language conversations with Khmer, Nepali, Vietnamese and Chinese women, frontline workers, and elders. For context and triangulation, we also collected stories through local and national ethnic news media sources. This shared process with multiple sources of story data enabled us to broaden and deepen our understanding of COVID experiences for Asian American communities beyond our own families.

### 3.2. Analysis of Story Data

Our group analysis of story data for *Hira Makes a Sound* began with K. Ty, Gurung, A. Ty, and Dương sharing images and themes from their sources. K. Ty mapped themes and connections. The team then compared the story data with our initial ideas, and we specifically discussed new insights gained from the research and documentation process, including intercultural nuances and intergenerational dynamics, conveyed to us in native languages. For example, while fully acknowledging the racist targeting of women and elders, our discussion and emerging storyline especially noted the strength and power of the participants' voices and stories. This shifted our recognition away from elders' status as vulnerable victims and toward their beloved stature as knowledge-holders, culture-bearers, and resilient survivors.

### 3.3. Story Development

As we pooled our wealth of content ideas and aesthetic visions for the product, the story emerged creatively and collaboratively. Having co-produced dozens of digital storytelling projects together as well as independently, including our own individual AsAmSt 370 digital stories, we wholeheartedly trust what and how every member of the team is sharing with each other. For example, co-author Gurung reported an experience involving a traditional ritual that enabled their family to regain spiritual strength to cope with the double pandemic's lasting effects. This inspired us to ground our fictional story, in part, on the experiences of our local Nepali American community, whose issues, voices, names, practices, and contributions are minimally represented among publicly accessible Asian American family and community narratives.

After a series of story brainstorming and outlining sessions, the team met with Tang to present a first draft of the story treatment. As the faculty mentor and executive producer of all storytelling projects, Tang provides direct, honest feedback for our work-in-progress, particularly in relation to thematic clarity, message consistency, and how best to convey complex cultural meanings in English-dominant text. Having an experienced and trusted veteran teacher or community leader, or narrative producer review the draft at this point is crucial. The most difficult aspects of a story's development—especially one that is collaborative, research-based, cross-cultural, and message-driven—is resolving critical questions, such as: (a) How to synthesize the creative ideas shared by everyone; (b) How to integrate important themes and perspectives from the research findings; (c) How to capture and convey the deep, complex meanings of non-English languages and cultures and ensure that those meanings are not lost in translation; (d) How to stay focused on the message of the story from beginning to end, in both the storybook content and the production process.

### 3.4. Storybook Production

Led by A. Ty, the storybook production process involved developing both text and illustrations for the final product. Being Khmer American rather than Nepali, A. Ty quickly recognized her limited familiarity with culturally specific references in the story, including names of the characters, relational statuses, meanings of spiritual traditions and objects, and more. Throughout the storybook production process, she had to communicate closely with other team members, especially co-author Gurung, to edit her written and visual representations of the story continually.

### 3.5. Translation

Based on the completed English version of the storybook, four experienced translators connected to the team began their own collaborative processes to produce bilingual versions of the text in Nepali, Khmer/Cambodian, Vietnamese, and Chinese languages. Each translator met with K. Ty and the linguistically relevant bilingual team member to validate and affirm the overall message of the storybook. The translators also met with Gurung to further clarify specific nuances of the story related to Nepali traditions and rituals. Bilingual team members also engaged their own elderly family members to brainstorm or confirm specific vocabulary used in their native languages. These specific layers of the process over several more months were time-consuming but essential to engage bilingual/bicultural team members fully and to respect the diversity and complexity of Asian cultural traditions and linguistic meanings.

### 3.6. Cultural Knowledge and Wisdom Access

Throughout these last three stages of the process—story development, storybook production, and translation—we also closely worked with a close family friend of co-author Gurung who is a former school principal in Nepal and a well-respected expert on the Gurung culture of Nepal and in the Nepali diaspora. He willingly provided invaluable knowledge and feedback to our team thanks to his affirmation of co-author Gurung's relational status as a trusted young-generation leader in the Nepali American community.

### 3.7. Sharing the Product

Following the Fall 2021 campus launch of a beta version of the storybook, a series of pilot readings took place in 2022 to generate feedback. Sample sites included a fifth-grade public elementary school classroom with predominantly Black and Latinx students in Boston, a gathering with Vietnamese bilingual K-12 educational leaders, professional development training for Massachusetts K-12 teachers on anti-racist curriculum development, and our own Asian American Studies courses. To encourage more impactful usage of the storybook in school, home, and community settings, we are also producing a concise Educator's Guide for *Hira Makes a Sound* that will highlight subthemes and details to explore, such as the role of Hira's mother as a frontline health care worker and the significance of the cultural community center, especially for the immigrant elders in the story. A glossary of Nepali relational terms will be provided, along with recommended resources for addressing anti-Asian hate.

### 3.8. Processing Reflections and Reflecting Process

With the completion of the beta version of the storybook in English, team members iteratively invested in an important cycle of active reflection from their own subject positions to document and synthesize their collective learning as Asian American Studies researchers, co-producers, storytellers, artists, and educators. Co-author K. Ty, who led the project, for example, shared several important insights:

> *"After gathering and compiling the oral history interviews from Asian elders in our respective communities, our team processed the themes that had emerged. One of the themes was the significance of alternative/spiritual/indigenous/ancestral knowledges that our elders drew on as sources of strength and meaning-making. Our team engaged in*

*a discussion on how these sources of knowledge were used by our elders to cope during a double pandemic of anti Asian racism and the global COVID-19 pandemic. We also recognized that our elders' forms of knowledge were typically unacknowledged in western knowledge paradigms and research practices... Growing up, my mom would pat my back or chest whenever I was frightened or when something too cold touched my body. She would tell me that she was calming my spirit so that it would not leave my body. I quickly understood the purpose and meaning of the shyayee shyayee ritual when [team member and co-author] Parmita [Gurung] explained it because of my own Khmer cultural frame of reference and lived experiences."*

While highlighting the importance of connections to community environments with spiritual purpose and power, co-author Dương similarly reflected:

*"Stories of fear, loneliness, uncertainty, health concerns of the coronavirus, their meaning of healing and recovery, and so many more came out of our experiences with having conversations with these grand aunties and aunties of our communities. There was one theme that stood strong and clear for all of us, it was a mix of spirituality and the ways of knowing how to survive that we as young Asian American women of today might not know how to fully contextualize, but the sense of connection and understanding came naturally to most of us on the project team. In my case, I grew up in a Vietnamese Buddhist household, I remember as a child going to temples with my grandmother. I participated in person on days of Dam Gio, also known as death anniversary. The Vietnamese families and communities reunite and celebrate the Day of the Dead by providing offerings to the dead. There is a great sense of honor and responsibility when dealing with spirituality in our communities... I grew up carrying these cultural knowledges and assets because of my firsthand experiences of spirituality despite not knowing how to articulate it in words. I remember the feelings of being in these cultural spaces, most of the time it took places in homes that harness great spiritual powers affecting my human senses, yet I couldn't fully describe it. The smell of the incense, the sounds of chanting, the air of the room and space, it was quite memorable for me. I felt a sense of nostalgia when Parmita spoke about the ritual, family and community members coming together to call the return of the wandering spirits. Immediately, I connected with what she was sharing with the team... I felt a connection with Parmita's details of the ritual. It reminded me of how my grandmother and family honor the spirits and souls. Afterwards, I felt the urge to tell the project team that we need to work the ritual part into the story treatment."*

Noting the rigorous care and responsibility required throughout the creative process with regard to visually depicting the main characters' family and community cultural environments in the story, co-author A. Ty added:

*"It took several conversations with the team and especially with Parmita to understand certain spiritual concepts like ether and the relationship between mantras, prayer flags and sounds. Sometimes it is still difficult for me to fully grasp or articulate the concepts. I think part of that has to do with the loss of certain meanings and concepts in translation from Nepalese to English. For example, the word ether is translated into a single word, sky, which can be fully understood on a different level in Nepalese but in English, the simplification of the translation to sky means that so much depth is lost. As co-author and illustrator, this made the creation process difficult... It took a lot of deep listening to Parmi's stories to feel what she might have felt and then to represent those feelings through illustrations. I also relied on my own culture's traditional practices that are different, but in some ways offer similar healing powers for my community, like coining which pushes bad air out of the body. Of course, it's not that similar, and I went through many revisions with Parmi's input to make sure that the illustrations were accurate. For example, I mistook the concept of mantras traveling through wind and sound by drawing the idea using wind chimes. Parmi corrected this by explaining that the mantras travel from prayer flags, not wind chimes. These details seem small but are very important and helped me gain a better understanding about the practices and spiritual concepts."*

Finally, co-author Gurung—whose rich cultural knowledge and community networks are so vividly acknowledged above as being crucial not only to the creation of *Hira Makes a Sound* but also to the shared learning of everyone in the project—reflected in a similarly personal, yet even more elemental way, given her distinctive subject position in relation to our process and product:

> *"I felt very emotional when I was reading the book in its entirety. I could look at the images and say which characters look like a real person in my life. I had known a Ratna Bajei my whole life, but I never knew what her name meant until we looked it up for this project. There is also a Neelam and Hira in her family, three different generations. When do you ever find something so real like this in a children's book?"*

After reaching the final stage of curated product-sharing and wider public dissemination, the focus of attention and analysis invariably shifts to the storybook itself and its independent impact in other educational environments. By highlighting the process, people, and platform responsible for its vision and co-production in this article, however, we stress the importance of the Asian American Studies environment where Hira was imagined and co-created. Within our own programmatic conditions and context where we experience energy as being both conserved and transformed, ethereal echoes from the sounds made by Hira and her community will continue to resonate across time and space. But like any empirical description of energy flow, power, positionality, or potential within a system, our description here comes from somewhere real that we are responsible for: our own environment.

## 4. Discussion

*Hira Makes a Sound* was researched, imagined, and co-produced as a purposeful storytelling intervention to meaningfully respond to the dual-pandemic realities of COVID-19 and anti-Asian violence. Following the March 2021 targeted murders of six Asian and Asian American women in metro Atlanta, many dozens of higher education institutional leaders throughout the country released statements of grief, outrage, and solidarity in relation to Asian American constituencies [37]. Some were heartfelt, others performative. Yet, few college leaders today are still referencing those expressions of support that they delivered with such urgency 20+ months ago. In contrast, references to *Hira Makes a Sound*, though modest, are still ongoing and growing.

Clearly, one important and continually under-valued educational role of Asian American Studies at all levels, from pre-K to post-doctorate, is to analyze historical and contemporary dynamics of anti-Asian violence in U.S. society. Despite our own efforts to do exactly that in AsAmSt courses every semester at UMass Boston for more than three decades, the effects have not been sufficient, in Hira's words, to "make a difference" in preventing the contemporary period of COVID-19 anti-Asian hate nationally or locally. Nevertheless, our own priorities continue to model creative alternatives through curricular innovation and classroom pedagogy with culturally sustaining themes of community resilience, responsibility, communication, and care. We are especially committed to projects designed with inter-generational, multilingual approaches that engage younger generation learners with Asian American elders in order to draw on, contribute to, and further invest in narrative generational wealth among all who participate.

While we are enthusiastically making plans to deploy Hira's story and readily envision a series of Gurung family sequels or fresh projects featuring characters from other under-represented communities, nevertheless, our primary purpose in this article is to identify the impacts, innovations, and inspirations made possible through our larger programmatic environment. In particular, by offering secure, course-based, curricular portals that have focused on Southeast Asian American student, family, and community contexts since 1989 and culturally specific Vietnamese American and Cambodian American Studies teaching and learning environments since 2003, we have modeled decades-long visions and practices that reach far beyond the basic, albeit necessary, national agenda to prioritize Southeast

Asian American data disaggregation as a baseline to achieve educational equity for under-resourced, under-represented, first-generation students with refugee family backgrounds.

Additionally, by developing and sustaining a curricular and pedagogical environment for student- and community-centered storytelling over two decades, we have enabled both a transformative learning process for diverse, urban, first-generation students and the co-production of ~400 narrative-based digital video and multimedia products that comprise a rich, ever-growing research archive documenting AANAPISI students' authentic contexts and experiences in both cross-sectional and longitudinal ways.

Through these and other decades-long examples, the programmatic environment of Asian American Studies has provided long-term commitments for:

- Facilitating socio-culturally responsive and academically relevant learning communities that support student persistence, mentoring, and connection at our urban, working-class, commuter university;
- Documenting significant issues, needs, and interventions in local Asian American communities and on campus, recognizing that our own students and alumni are themselves members and participants within local neighborhoods, workplaces, and community-based institutions;
- Building research and development capacities in local Asian American communities through connecting ethnic studies perspectives, interdisciplinary methodologies, and analytic frameworks with students' and alumni's diasporic social networks and cultural/linguistic knowledge;
- Producing and preserving original collections of locally relevant source materials, including digital stories and original children's storybooks, as well as oral histories, literary anthologies, photo essays, and other archival resources.

Our Asian American Studies environment can be understood and described in multiple ways: the physical spaces, people, and trusted relationships; the well-disciplined curricular and pedagogical practices; the multiple skills, languages, cultures, perspectives, stories, and shared visions; the academic course structures and secure course schedules sustained over many years; the overarching commitments to capacity-building, knowledge production, and educational transformation for students and communities, etc. We have consistently demonstrated that the development of long-term ecologies to support culturally sustaining curricula and storytelling co-production processes strategically led by core faculty with students and alumni are not just conceptually compelling but pragmatically possible. These dimensions of transformative Asian American Studies education are much-needed but rarely considered in assessments of equity or inclusion within higher education institutions [38].

## 5. Conclusions

The complex, structural, and cultural processes of creating and sustaining more equitable campus environments for first-generation Asian American students in college—especially those from low-income, immigrant/refugee families and communities attending the critical sector of minority-serving higher education institutions designated as Asian American and Native American Pacific Islander-Serving Institutions (AANAPISIs)—are under-studied, but increasingly gaining attention [1,3,39]. Federal AANAPISI designation implicates specific local family and community contexts that drive a critical mass of Asian American or Pacific Islander students to attend college. Simply qualifying demographically for AANAPISI status, however, says little about the qualities or impacts of specific campus environments where students have opportunities to experience inclusive, equitable, transformative education [40,41].

We argue, therefore, that those AANAPISIs with longstanding Asian American Studies curricular and pedagogical commitments to such transformative educational practices can offer especially valuable insights for U.S. higher education [5,6,42]. Faculty-led classroom and curricular environments, whether in Asian American Studies or other related, relevant fields, are notable for their potential to be independently sustained from either grant-

dependent, time-limited, student-success interventions or professional staff infrastructures within student affairs units. This academic program-based environmental strategy is particularly salient for under-resourced, predominantly commuter institutions where student engagement with faculty and peers in classroom environments is precious and primary.

In our specific case as an urban public AANAPISI research university with a 35-year programmatic commitment to develop and lead an educationally transformative Asian American Studies teaching and learning environment, we have sought to demonstrate, document, and model innovative, high-impact practices that are not only possible to pilot, but also to sustain.

In our most recent dual-pandemic storybook example of *Hira Makes a Sound*, Hira's grandmother, Ratna Bajei, is hospitalized briefly after being assaulted on the street because of anti-Asian hate. Though she is released from the hospital after the medical doctor's examination finds her to be "recovered," she is no longer fully herself after the attack. As Hira learns, for Ratna Bajei's spirit to return home, she needs to culturally and spiritually reconnect with the unique sounds and collective, multigenerational energies emanating from the Lali Gurans Community Center, where her Nepali diasporic family and friends are intentionally concentrating. To be whole, she needs both the familiar grounding and ethereal inspiration that are attentively co-produced and holistically sustained within that beloved environment.

Metaphorically, we well-recognize and continually aspire to provide a comparably equitable and inclusive environmental context for ourselves and all who join with us in Asian American Studies at our urban public AANAPISI research university. This decades-long commitment is more necessary than ever during our dual-pandemic times in the 2020s.

**Author Contributions:** Conceptualization, P.N.K., S.S.T., K.S.T., P.G., A.T. and N.D.; methodology, S.S.T.; investigation, K.S.T., P.G., A.T. and N.D.; data curation, K.S.T. and A.T.; writing—original draft preparation, P.N.K., S.S.T., K.S.T., P.G., A.T. and N.D.; writing—review and editing, P.N.K., S.S.T., K.S.T., P.G., A.T. and N.D.; supervision, S.S.T.; project administration, K.S.T.; funding acquisition, P.N.K., S.S.T. and K.S.T. All authors have read and agreed to the published version of the manuscript.

**Funding:** This work was funded, in part, by the Racial Justice Fund of the College of Education and Human Development at the University of Massachusetts Boston and by the Asian American and Native American Pacific Islander-Serving Institution (AANAPISI) Program of the U.S. Department of Education, CFDA 84.382B, grant number P382B160017.

**Institutional Review Board Statement:** There was no IRB involvement because the content of the manuscript describes and reflects on teaching and learning commitments carried out over 2–3 decades of time rather than a discrete research project involving research subjects. We are writing about work we are directly responsible for leading and creating from our stance as the teachers and program staff. We are not presenting results or findings that argue for reproducibility or generalizability in the sense of scientific or IRB-relevant research. One section of the manuscript (3.1) briefly mentions our holding conversations with family and community members about the dual pandemic as part of context-setting for the storybook development process. These conversations were considered comparable to our use of oral histories in storytelling projects, and are exempt from IRB oversight. Furthermore, the product being described in that section of the article is a fictional storybook.

**Informed Consent Statement:** Not applicable.

**Data Availability Statement:** Not applicable.

**Conflicts of Interest:** The authors declare no conflict of interest.

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
