# Peer review of "Hira Makes a Sound: Sustaining High-Impact AANAPISI Innovation in an Asian American Studies Environment before and beyond the COVID-19 Anti-Asian Hate Pandemic"

_education, doi:10.3390/educsci13020128_

Round 1

Reviewer 1 Report

Thank you for the opportunity to review this manuscript.  In all of my reviews, never have I read a manuscript that kept me gripping for the next sentence, to learn more about the amazing work being co-constructed at UMass Boston.  I offer very few notes for revisions, and know that this piece will contribute greatly to the literature.  It is, simply put, inspiring. 

In the 2.1.2. Vietnamese American Studies Commitments section, it would be wonderful to learn a bit more about the community in Dorchester.  I think situating that bit of context, as was done for the Cambodian American section, might let readers who may be unfamiliar with Boston gain some additional context. 

I loved the discussion of the utility of disaggregated data.  No comment on revisions, just grateful for this important concept that must be discussed when we write and demand data disaggregation. 

Page 6, line 256, the = should be a – for first-generation

Page 10, line 247, I suggest that the authors refer to the 6 victims in Atlanta as “Asian American women” rather than “Asian women”.  Given that they lived, worked, had families in the United States, I would say they are Asian American, rather than Asian.  I know the media refers to them as Asian, since they may not all have maintained US citizenship, but I don’t believe that should be the litmus test used to determine if one is Asian American or Asian. 

Finally, at the start or perhaps at the end, I think the authors should speak to the uniqueness of UMass Boston’s Asian American Studies program.  It is so distinct from AAS programs found at so many other institutions, where frankly, those programs appear to have limited, if any, connection to community.  Indeed, scholars have critiqued the evolution of the field from its roots.  But UMass Boston continues to maintain these roots, and from there grow a forest of trees to push the field in new and meaningful trajectories that bridge the “academy” with community

Author Response

Reviewer #1: 

Thank you very much for your thoughtful and encouraging review. We so appreciate your direct understanding/connecting with our intentions and the draft of our content. In response to your suggestions and specific requests:

We provided additional background context regarding Boston’s Vietnamese community as requested in 2.1.2.

We used “Asian and Asian American” in reference to the six women murdered in metro Atlanta [rather than replace”Asian” with “Asian American”], deciding that both was more appropriate under the circumstances that one or the other alone.

We provided some additional clarification about the unique qualities of UMass Boston’s Asian American Studies Program, as requested, mainly in the Discussion section.  

We corrected “first-generation” typo as noted.

Reviewer 2 Report

The authors argued a compelling case about the unique and frankly vital role that AANAPISIs with established Asian American Studies programs play in the landscape of American higher education. It appears that the authors themselves reflect the intergenerational intentionally that the AsAmSt program at UMassBoston approaches their curricula and pedagogy. I commend the authors for this undertaking. In their piece, the authors weave together the stories of AANAPISIs and Asian American Studies – one may assume that AANAPISIs somehow already have culturally relevant curricula, but those who study higher education, Asian Americans, or related contexts would know that this is far from the case. As such, this piece continues to push the conversation that before AANAPISIs existed, there were people, programs, structures that served AAPI students – and that it would be important for AANAPISI scholars and policymakers to recognize this fact and inform their work.

I often find it uneasy to accept masked peer reviews where the reviewers provide mostly constructive feedback without situating from where the feedback comes. To disrupt this power dynamic a bit, I offer a bit of my own positionality. I am a first-generation faculty member in education who also has a research and practice interest in AANAPISIs. I am also a product of ethnic studies and urban education. As an educator, a woman of color, I strive to similarly see and construct a classroom of possibility for those students whom meaningful connections and learning are vital to their success. It is from these salient lenses and identities that I provide my comments below to strengthen the manuscript.

There are only a few comments I have, as the manuscript is already quite strong.

Your manuscript does a fantastic job painting the longstanding and deep foundation supporting curricular and programmatic innovations. I’d encourage the authors to attend to the amount of attention paid to laying out these two foundations compared to the production/impact of the storybook. Currently, five pages are devoted to the foundations and about three to the storybook. I share this because reviewers often comment on what to add but not what to refine to make space for the additions. There are two additions I encourage the authors to make:

The first is considering the through-line related to COVID-19/anti-Asian hate pandemic. As suggested by the title, the AsAmSt environment existed before and beyond the pandemic(s). Yet, both anti-Asian hate and the COVID-19 pandemic are only mentioned in section 3. The through-line would be strengthened if in section 2 (about the foundations), you were able to include/lift up the ways in which these foundations countered anti-Asian hate (which we know existed before COVID-19). One may assume that the mere existence of AsAmSt is a counter to anti-Asian hate and violence; yet, as students and scholars of ethnic studies, we understand that we cannot assume others know or understand this fact. As such, if there are ways that AsAmSt combated anti-Asian bias/hate/discrimination (individually, institutionally, etc.), it may draw a stronger connection with WHY such foundations offer the best environments to combat present-day anti-Asian hate.

The second addition is the impact of engaging in the production of Hira Makes a Sound on co-authors 3-6. Section 2 (foundations) tells the reader that these spaces are transformative; yet, Section 3 is primarily about what happened and how the authors worked together to produce the storybook. This leaves me thirsting for how participating and engaging in this production impacted/shaped the authors 3-6. In what ways did engaging in this production offer an avenue to channel complex emotions/feelings/experiences about the in-your-face anti-Asian hate that surged/is surging along with the COVID-19 pandemic? What connections can authors 3-6 make between this experience of producing the storybook with their own learnings (in and out of the classroom)? In short, providing these reflections will not just TELL the readers that these spaces are transformative but SHOW the readers how they do so.

A final major comment for the authors to consider is the distinction between the discussion and conclusion sections. Some of the content in the conclusion could be in the discussion (lines 528 through 556, for example, read more appropriate for the discussion section). This may help the authors focus on drawing out why such long-term commitments are even more necessary during the current pandemics (and also strengthen the through-line – my comment above).

A few minor edits for the authors to attend to:

-       Throughout the manuscript, correct where the “and” is in the phrase “Asian American AND Native American Pacific Islander Serving Institution(s)”. The designation is two groups “Asian American” and “Native American Pacific Islander.” See: https://www2.ed.gov/programs/aanapi/index.html

-       Correct hyphen in phrase “first=generation” line 257

-       Reconsider the use of the term “womxn” as there are calls from trans women, particular transwomen of color, to stop using this term – in short, because trans women are women. Although the intention of “womxn” may have been to be inclusive, it often has the impact of centering cis women who include trans women, thereby, imposing a conditionality on trans women. See this article: https://www.subvrtmag.com/why-womxn-harmful-term-trans-non-binary-femmes/

-       Consider engaging pieces from an About Campus special issue on AANAPISIs in your manuscript (https://journals.sagepub.com/toc/acaa/26/1). Specific articles include:

o   Wang, Mac, and Museus’ piece on ethnic studies at AANAPISIs in your manuscript

o   Daus-Magbual’s piece on ASPIRE SFSU

Author Response

Reviewer #2:

Thank you very much for your thoughtful and encouraging review. We so appreciate your direct understanding/connecting with our intentions and the draft of our content as well as your explicit sharing of your own subject-position in relation to this work. In response to your suggestions and specific requests:

We addressed the observed imbalance of greater attention/content on the curricular innovations and less on the storybook intervention by adding a significant 3.8 subsection with reflections from team members regarding their own learning through the project process/product as requested. 

In 2.1.1., we clarified/added examples of addressing anti-Asian hate/violence historically in the courses long before the COVID-19 period, thereby strengthening that through-line as requested.

We removed some content from the conclusion and moved it into the discussion section specifically as requested, along with a transitional sentence to support the move.

We included the suggested references from the About College special issue on AANAPISIs which were very appropriate.

We corrected AANAPISI and first-generation typos and edited “womxn” to “women” as noted.

Reviewer 3 Report

Dear Authors,

I had the pleasure of reviewing the manuscript, “Hira Makes a Sound: Sustaining High-Impact AANAPISI Innovation in an Asian American Studies environment Before and Beyond the COVID-19 Anti-Asian Hate Pandemic.” I thank the authors for their time and effort in collaborating on this piece and contributing important perspectives about the value and importance of Asian American Studies pedagogy and curriculum. I believe this piece warrants publication and would look forward to reviewing and citing it in the future. I have some comments below that I hope are useful.

I have a great appreciation for this piece regarding the point about calls for data disaggregation being incomplete without a deep understanding of how curricular and pedagogical environments have already provided “an answer” to what should be done. While calls for disaggregation are important and necessary, at what point do we lose focus and prioritize disaggregation without an understanding of “what next” and how to address the issue that we believe disaggregation will solve? Especially the point about disaggregation not changing systems and structures is an apt one. Also, the fact that some Asian American Studies spaces have always had pedagogical commitments around advocacy is an important point to remember. There are also lessons for future AANAPISI-status seekers about the critical role that Asian American Studies programs SHOULD play in the process. We have much to learn from spaces like UMass Boston’s Asian American Studies program about how to build connections with students and communities through a pedagogy that centers knowledge production from those within these communities.

I noticed various versions of AANAPISI being used. I encourage a review of these and to correct it to the version noted in federal policy: “Asian American and Native American Pacific Islander Serving Institution.”

How long did the process of developing the book take? There were mentions of “longer-term research and documentation” and “time-consuming” and a very thorough description of the various elements and conditions that shaped the development of the book but it was unclear what the timeline was for its creation.

Overall, the piece was engaging and made significant points that will shape my own thinking about the important role Asian American Studies plays in advocacy for populations such as SEAA and immigrant and refugee communities. The biggest area of recommendation is that the discussion and conclusion are out of balance, with the conclusion being significantly long. Much of what was written in the conclusion could be incorporated into the discussion. For example, the paragraph on “we argue, that those AANAPISIs with longstanding Asian American Studies…” could be better suited in the discussion section (and or a different implications section). Also, the entire section starting with “Additionally, by developing and sustaining a curricular and pedagogical environment…” could also be shifted to the discussion or implications section. 

Thank you for your work and deep commitment to working with SEAA, immigrant, and refugee communities over time. 

Author Response

Reviewer #3: 

Thank you very much for your thoughtful and encouraging review. We so appreciate your direct understanding/connecting with our intentions and the draft of our content. In response to your suggestions and specific requests:

We corrected AANAPISI typos as noted.

We clarified some aspects of the 18-month timeline associated with the storybook project as requested.

We removed some content from the conclusion and moved it into the discussion section specifically as requested, along with a transitional sentence to support the move.